# Optimal Population Coding for Dynamic Input by Nonequilibrium Networks

**DOI:** 10.3390/e24050598

**Published:** 2022-04-25

**Authors:** Kevin S. Chen

**Affiliations:** Princeton Neuroscience Institute, Princeton University, Princeton, NJ 08544, USA; kschen@princeton.edu

**Keywords:** efficient coding, population coding, nonequilibrium, kinetic ising model

## Abstract

The efficient coding hypothesis states that neural response should maximize its information about the external input. Theoretical studies focus on optimal response in single neuron and population code in networks with weak pairwise interactions. However, more biological settings with asymmetric connectivity and the encoding for dynamical stimuli have not been well-characterized. Here, we study the collective response in a kinetic Ising model that encodes the dynamic input. We apply gradient-based method and mean-field approximation to reconstruct networks given the neural code that encodes dynamic input patterns. We measure network asymmetry, decoding performance, and entropy production from networks that generate optimal population code. We analyze how stimulus correlation, time scale, and reliability of the network affect optimal encoding networks. Specifically, we find network dynamics altered by statistics of the dynamic input, identify stimulus encoding strategies, and show optimal effective temperature in the asymmetric networks. We further discuss how this approach connects to the Bayesian framework and continuous recurrent neural networks. Together, these results bridge concepts of nonequilibrium physics with the analyses of dynamics and coding in networks.

## 1. Introduction

A fundamental issue in neuroscience is neural coding, which characterizes how neural activity represents external input. Efficient coding is the classical hypothesis stating that neural activity should maximize its information about the stimuli it receives [1,2]. Maximization of mutual information between sensory stimuli and spiking patterns has been tested experimentally and studied in neural models with respect to certain constraints [1,3,4,5]. However, prior literature mostly focuses on single neuron coding, non-interacting population, or with static stimuli. It is less clear how population coding in a network should be optimized for dynamic input [6,7,8]. Here, we study the optimal population code for dynamic input with a simplified neural network model, the kinetic Ising model, and analyze how dynamics and structure of these optimal networks are affected by the encoded signal. With this simple model and least assumption for the biophysical mechanisms, we aim to explore the relation between network encoding of dynamical signals and its nonequilibrium properties.

There are two main frameworks developed in the past two decades for describing neural population activities: the generalized linear model (GLM) [2,9] and maximum-entropy method (MaxEnt) [10,11,12]. GLM provides a statistical framework to capture how neural responses are affected by external stimuli, its own spiking history, and contribution from other coupled neurons. This method successfully captures temporal dependency to dynamic stimuli as well as the correlation structure across neural population. Moreover, GLM can be extended for stimulus decoding in a Bayesian framework. One can estimate the external input given the neural response and further approximate the posterior distribution given neural response to quantify information transfer. A potential downside of this model is the instability of certain dynamics or in regimes with strong interactions. Recent studies focus on methods to mitigate these problems and also extend the framework to dimension reduction and inferring latent dynamic [13].

MaxEnt method was motivated by statistical physics and aims to capture the collective response in the neural population with least model assumptions [10]. The model uses average activity of each neuron and the pairwise correlation as constraints and learns the parameter that maximizes entropy at thermal equilibrium. MaxEnt successfully predicts higher-order interactions, reveals the energy landscape of neural code words, and further suggests error-robust population code organization with real neural recordings [14]. This method relies on the assumption of an equilibrium distribution of activity pattern and ignores the temporal structure. Follow-up studies extend the method to capture MaxEnt of spatiotemporal activity “blocks” through time or conditioned on stimulus [11,15].

There remain complementary gaps between these methods of studying population coding. The link between GLM and more general forms of recurrent neural network models is not obvious [16,17,18,19]. For MaxEnt, the imposed assumption for symmetric function connectivity and equilibrium neural dynamics are often not suitable for more realistic biological networks [20,21,22]. We aim to integrate these frameworks by using Bernoulli GLM as the stimulus encoder, constructing a network with an asymmetric Ising model that produces such spiking patterns, then further decoding the external input from the network activity. This framework incorporates network interaction similar to the MaxEnt method and stimulus encoding in the GLM framework [2,13,23].

There is an increased interest in studying nonequilibrium network dynamics [21,22,24,25]. The study of network dynamics and inference methods under nonequilibrium settings is crucial for biological networks [26,27]. More specifically, kinetic Ising model is the generalization of Ising models out of equilibrium. Rather than modeling the equilibrium distribution in Ising models, the kinetic Ising model can describe state transition probability with asynchronous updates and asymmetric connectivity. In terms of statistical models, kinetic Ising model is similar to GLM as it captures the conditional probability given history spikes, but the exact mapping is not obvious. Inference for this model ranges from Boltzmann learning, belief-propagation, to mean field approximation [27,28,29]. Numerical methods show that one can successfully reconstruct parameters given non-stationary spin dynamics from the kinetic process. We utilize these methods to study kinetic Ising models with target spike trains.

To study the neural encoding of dynamical signals in an interacting network, we focus on a kinetic Ising model driven by dynamic input. We compute the optimal population code for such dynamic input and train the network model to generate the target spikes for decoding [30,31]. The key finding is that the optimal network structure depends on statistics of the dynamic input and the reliability of neural response. This suggests that a network optimally encoding dynamical signals should adapt to or learn from the signal statistics. The results provide testable predictions relating structural measurements to the encoded stimulus, as well as the design principle for dynamical population code in a network. Finally, we draw links between the nonequilibrium thermodynamics of neural networks, different neural coding strategies, and generalized forms of recurrent network dynamics.

## 2. Model and Analysis

We review the neural encoding and stimulus decoding framework. We then present the nonequilibrium network model, which is known as the kinetic Ising model, and formulate the corresponding inference method. The idea is to train a kinetic Ising model to produce output that optimally reconstructs the external input. We analyze decoding performance and network structures from these optimal models under different dynamic input or parameter settings.

### 2.1. Neural Encoding and Decoding Framework

We assume that the external input can be linearly reconstructed from neural activity, namely a linear decoding framework [16,17]. The schematic and example traces are shown in Figure 1. The estimated input follows:(1)$$\hat{x}\left(t\right)=\Phi^{T}r\left(t\right)$$
where Φ projects the response vector $r\left(t\right)\in R^{N}$ at time *t* to construct the stimulus vector noted as $\hat{x}\left(t\right)\in R^{d}$. The matrix Φ acts as a decoder that projects signal from the network dimension to the stimulus dimension. Here we construct $\Phi \in R^{N\times d}$ with random Gaussian elements from N(0,1). The probability distribution of this reconstruction $\hat{x}$ is approximated with Gaussian:(2)$$P\left(x\left(t\right)|r\left(t\right)\right)∼N\left(\Phi^{T}r\left(t\right),\Sigma \right)$$
where Σ is the variance around the mean reconstruction. For convenience in deriving the framework of generating neural activity given target patterns, we drop the time notation *t* temporarily and focus on variables at a given time point. The stimulus decoding probability in Equation (Equation 2) can be thought of as a Gaussian approximation of the posterior distribution in a Bayesian framework: P(x|r)∝P(r|x)P(x), where P(r|x) is the neural encoding model that produces activity pattern *r* given external input *x* and P(x) is the prior for input [9]. More specifically, here the posterior mean is $\Phi^{T}r$ and the Hessian of log posterior is $\Sigma^{-1}$. We construct the generative process for neural response from an optimal encoding model: P(r|x)∝P(x|r)P(r). This is a form inspired by the “neural engine” framework [23] that has an Ising model prior with binary activity. One tractable prior on rate activity would be a multivariate Gaussian distribution $P\left(r\right)=\frac{1}{Z}\exp\left(r^{T}\Omega r\right)$, where *Z* is the normalization constant and $\Omega \in R^{N\times N}$ is the pairwise interaction matrix [23,32]. The posterior for generating neural response is now:(3)$$P\left(r|x\right)\propto \exp\left(\frac{1}{2}r^{T}\Omega r+x^{T}\Phi \Sigma^{-1}r-\frac{1}{2}x^{T}\Phi^{T}\Sigma^{-1}\Phi x\right)$$
where first term corresponds to the prior on activity, the second term reflects an effective correlation between input and activity, and the third term is the covariance of input signal projected to the network. Here, our aim is to connect the linear decoding model with a rate variable with the binary spiking activity we would use in the encoding network. The general framework allows the exploration of different assumptions of the network activity prior, with diagonal terms as bias and off-diagonal terms as pair-wise interactions of neural activity in Ω matrix. For simplicity, here we use an empirical prior for activity r. This can later be shown in the network encoding model Equation (Equation 7) with the an uncorrelated initial network structure before learning. Given a random network connectivity, the neural activity is close to uncorrelated, corresponding to an identity matrix for Ω. By taking the derivative of Equation (Equation 3) with respect to r, we find the optimal posterior response mode $r=-\Sigma^{-1}\Phi^{T}x$. We use this mode of spiking rate response, now denoted as λ, to generate spike trains by passing it through Bernoulli process with logistic nonlinearity known in canonical neural encoding models [9]. Applying this spike generation process through time:(4)$$s_{i}\left(t\right)∼Bernoulli\left(\frac{1}{1+\exp(\lambda_{i}dt)}\right)$$
where $s_{i}\left(t\right)$ is the binary spikes of neuron *i* in time bin dt, $\lambda_{i}\left(t\right)$ is the encoding spiking rate derived from Equation (Equation 3) to generate discrete spikes. When time step dt is sufficiently small, the result is close to spike trains from Poisson process.

Putting it all together according to Markov models, the population activity decodes time-dependent stimuli $P(x_{t}|s_{t})$ and the transition probability of spiking patterns $P(s_{t}|s_{t-1})$ would be further formulated below (this is a hidden Markov model [33] if the network activity *s* is unobserved). The goal is to find the optimal parameters through maximum likelihood estimation of this Markov chain. Given the goal to optimally encoding and decode stimulus *x*, we write down the data likelihood as $L\left(\theta \right)=\prod_{t}^{T}P\left(x_{t},s_{t}|\theta \right)$. The inference procedure is updating parameter θ with maximizing the log-likelihood:(5)$$\theta^{*}=argmax_{\theta }\Sigma_{t=1}^{T}\log P\left(x_{t}|s_{t}\right)P\left(s_{t}|s_{t-1},\theta \right)$$
where the optimal stimulus decoding term $P(x_{t}|s_{t})$ is fixed in our framework and is independent to the network parameters θ of interest, governed by stimulus decoding in Equation (Equation 2) and spike generation in Equation (Equation 4). The probability $P(s_{t}|s_{t-1},\theta )$ would be modeled with a nonequilibrium network discussed in the next section.

Another view of this decoding framework connects to the variational method for information optimization [26]:(6)$$I\left(x,s\right)=H\left(x\right)-\langle H\left(x|s\right)\rangle \geq H\left(x\right)-{\langle -\log P\left(x|s\right)\rangle }_{P(x,s)}$$
where H(x) is the entropy of the random variable *x*. The entropy of stimulus *x* is not subject to optimization, so we focus on the entropy of the decoding probability. The lower bound is on the right side of the inequality and the second term is substituted with the expected approximated log posterior. This expectation is computed for the joint probability. In other words, our attempt to search for the optimal decoding structure is equivalent to information optimization through maximizing the lower bound.

### 2.2. Nonequilibrium Network and Inference Methods

We use the kinetic Ising model to simulate nonequilibrium neural network dynamics [25,27,28]. The neural activities are vectors $s\in R^{N}$ with binary spins $s_{i}=\left\{+1,-1\right\}$ and the discrete time update rule follows:(7)$$P\left(s\left(t+1\right)|s\left(t\right)\right)=\Pi_{i}^{N}\frac{\exp(\beta s_{i}\left(t+1\right)H_{i}\left(t\right))}{2\cosh(\beta H_{i}\left(t\right))}$$
where β is the inverse temperature and the term $H_{i}\left(t\right)$ follows:(8)$$H_{i}\left(t\right)=\Sigma_{k}\Phi_{ik}x_{k}\left(t\right)+\Sigma_{j}J_{ij}s_{j}\left(t\right)$$
where Φ is the same projection matrix or kernel for stimulus x and $J\in R^{N\times N}$ is the connectivity within the neural network. Note that this potential term $H_{i}\left(t\right)$ depends on stimulus x and here Φ acts as an encoder that projects the input signal to the network space. Using another random matrix with the same dimension does not change our results. We use the transpose of the decoder matrix in Equation (Equation 1) to formulate the setup that is similar to an autoencoder shown in Figure 1. If Φ is an orthonormal matrix, the result would be trivial when there is no noise, and the optimal network *J* would be an identity matrix. However, this is not the case in our setting with finite temperature in the nonequilibrium network. In contrast to the typical Ising model, note that there are no symmetry constraints on matrix *J*. One can view the term *H* as an effective current subthreshold to the binary nonlinearity.

We generate spike trains for optimal linear decoding given the projection matrix Φ and covariance matrix Σ. With the target activity pattern, we learn the network parameters that generate such patterns under external stimuli. This is known as the network inference problem [12]. We introduce the gradient descending method to minimize the negative log probability of encoding model [34], which gives the learning rule:(9)$$\delta J_{ij}\propto \gamma \beta \left({\langle s_{i}\left(t+1\right)s_{j}\left(t\right)\rangle }_{t}-{\langle \tanh\left(\beta H_{i}\left(t\right)\right)s_{j}\left(t\right)\rangle }_{t}\right)$$
(10)$$\delta H_{i}\propto \gamma \beta \left({\langle s_{i}\left(t+1\right)\rangle }_{t}-{\langle \tanh\left(\beta H_{i}\left(t\right)\right)\rangle }_{t}\right)$$
where Equation (Equation 9) is the update process for connectivity *J* and Equation (Equation 10) is the update for effective field *H* that contains the input stimuli *x* (Equation (Equation 8)) and γ is the learning rate. We denote the reconstructed network structure as $\hat{J}$. One can decode the stimuli x given these parameters by solving Equation (Equation 8). Specifically, this learning algorithm for asynchronous kinetic Ising model is derived from maximum likelihood of the spin update probability, known as the “spin- and update-history-based” algorithm with an objective function $L=\Sigma_{i}\Sigma_{t}\left[s_{i}\left(t+1\right)H_{i}\left(t\right)-\log\left(2\cosh H_{i}\left(t\right)\right)\right]$ [34]. The form is similar to spike-time dependent plasticity known in neural learning rules, as it takes the delayed spike correlation into account. This is further explored in the discussion section.

Another method is the mean field approximation that is shown to be more efficient than gradient methods [27,28,35]. The estimated connectivity ${\hat{J}}_{MF}$ is computed through:(11)$${\hat{J}}_{MF}=B{\left(AC\right)}^{-1}$$
where $C_{ij}={\langle s_{i}\left(t\right)s_{j}\left(t\right)\rangle }_{t}$ is the correlation matrix and $B_{ij}={\langle s_{i}\left(t+1\right)s_{j}\left(t\right)\rangle }_{t}$ is the time delayed correlation matrix. Matrix *A* is a diagonal matrix with $A_{ij}=a_{i}\delta_{ij}$ with $a_{i}=\beta \left(1-m_{i}^{2}\right)$ and $m_{i}=\langle s_{i}\rangle$ is the mean activity. This is the naive mean field approximation that only keeps the first order term. Methods such as Thouless-Anderson-Palmer (TAP) mean field approximation with second-order terms and Plefka expansion with higher-order has been discussed in past literature [35,36,37].

### 2.3. Protocol of Network Simulation and Training

We generate dynamical stimuli with a linear stochastic equation:(12)$$\frac{dx}{dt}=Mx+\xi$$
where matrix $M\in R^{d\times d}$ governs the linear dynamics of vector x and ξ is the noise term $\xi ∼N(0,\Sigma_{x})$. The time series is used for encoding and decoding in the network. Another paradigm is used to study computation for state-dependent dynamics:(13)$$\frac{dx}{dt}=Mx+c+\xi$$
where *c* is another time series of input control signal to the dynamical system. The target is to reconstruct variable x with nonlinear dynamics under switching control *c*.

The analysis procedure follows: (1) We generate stimulus with parameters $\left\{M,\Sigma_{x}\right\}$ controlling the stimulus statistics and {Φ,Σ} for the aimed decoding signal (Equations (Equation 12) and (Equation 13)). (2) Given these variables, we generate time series of the stimuli for decoding and target optimal spiking patterns (Equations (Equation 1) and (Equation 4)). (3) These spike trains are then used for network inference to learn the optimal network parameterized with {J,H} (Equations (Equation 9) and (Equation 10)). (4) Finally, we analyze the optimal network parameter and population code (simulated with learned parameters using Equations (Equation 7) and (Equation 8)) with measurements described in the next section (Equations (Equation 14)–(Equation 17)).

Unless otherwise mentioned, we fix the following parameters for network dynamics and stimulus generation: N=10 neurons, T=10,000 time steps, β=0.5 inverse temperature, d=3 input dimension, Φ projection matrix generated from zero mean Gaussian distribution, Σ is an identity matrix, and network *J* is initialized with the inverse covariance of target activity s. For dynamical stimuli, $M_{ii}=0.5$ for the diagonal elements, $\Sigma_{x}=0.1$ noise intensity along the diagonal, and input dynamics normalized to be bounded {−5,5}. For inference procedure, we use 1000 iteration for the gradient descending method with γ=0.1 and 500 maximum iterations for convergence with the mean field methods. We investigate effects of stimulus statistics and the network reliability β. Other parameters such as simulation time and network size do not significantly change the results.

### 2.4. Analysis of the Optimal Population Coding

Given the network is trained to produce optimal neural code, we analyze the population code s(t) and optimal network structure $J^{*}$ for stimulus decoding with three main measurements: decoding performance *D*, network asymmetry η, and entropy production EP. The decoding performance is measured by the correlation coefficient between reconstructed stimuli and the true stimuli, and this is done across all stimulus dimensions in x:(14)$$D=\frac{Cov(x,\hat{x})}{\sqrt{Var\left(x\right)Var\left(\hat{x}\right)}}$$

The degree of asymmetry of matrix *J* is computed by the ratio of matrix norm between the symmetric (SY) and asymmetric (AS) components:(15)$$\eta =\frac{∥AS∥}{∥SY∥}=\frac{∥0.5\left(J-J^{T}\right)∥}{∥0.5\left(J+J^{T}\right)∥}$$

Lastly, entropy production EP along the generated neural trajectories is computed with a common form for irreversibility:(16)$$EP=\log(\frac{P_{ss}\left(\epsilon_{i}\right)T\left(\epsilon_{j}|\epsilon_{i}\right)}{P_{ss}\left(\epsilon_{j}\right)T\left(\epsilon_{i}|\epsilon_{j}\right)})$$
where $P_{ss}$ is the steady-state measurement of probability observing neural state ϵ and matrix T(x|y) is the transition probability from state *y* to *x*. This measurement reflects the violation of detailed-balance when it is nonzero, quantifying how far away the system is away from equilibrium [35,38]. In the setting of kinetic Ising model at steady-state, it is shown that the steady-state entropy production can be calculated with [35]:(17)$$EP=\Sigma \left(J_{ij}-J_{ji}\right)B_{ij}$$
where $B_{ij}$ is the time delayed correlation matrix in Equation (Equation 11) defined for the mean field method. This formula indicates that the entropy production reflects the asymmetry of the kinetic Ising model. We explore how this value changes for different stimulus and network constraints.

## 3. Results

### 3.1. Model Setup and Network Inference

The target population spikes were generated through the optimal decoding framework described in the previous section. The temporal and population correlation structure of the target spikes are governed by stimulus statistics and we characterize the corresponding network structure. Note that Bayesian decoding sets an upper bound for the performance and an untrained random neural network is introduced as a null model for comparison. We start by verifying the inference procedure, then applying the method to spiking patterns that encode the dynamical stimuli in the next sections.

The inference method recovers network connectivity given spiking activity (Figure 2). Specifically, the gradient-based method produces reasonable correspondence to ground truth connectivity. We show that the log-likelihood converges after training for over hundreds of repetitions (Figure 2a,b). The naive mean field approximation works within a parameter range, but fails for input driven networks near inverse temperature β=1 (Figure 2c). Inference procedure enables reconstruction of the input stimuli (Figure 2d). The higher variability and bias in reconstruction under dynamic input can potentially be due to the unidentifiable activity pattern produced when the network is driven by signal with an effectively lower dimension. This makes the network parameter less constrained by activity, so the network output can reconstruct the stimulus pattern *X* even when estimation of network parameter $\hat{J}$ seems noisier. The correlation coefficient of input decoding is higher for larger networks, but we still show a reasonable stimulus decoding performance with N=10 that outperforms model with random connectivity. As explored in the next sections, the decoding performance and network structure depend on input dynamics.

### 3.2. Effects of Dynamic Input

We alter the correlation of stimulus time series applied to neurons in the network by tuning off-diagonal values of matrix *M* in the stochastic Equation (Equation 12). Here we focus on a two dimensional (d=2) stimulus x, so the correlation is $\sigma ={\langle x_{1}x_{2}\rangle }_{t}$. The results show that the asymmetry of the network η increases when the stimulus correlation across neurons increases (Figure 3a). In other words, when the external input has lower effective dimension, the optimal network has larger asymmetry. Entropy production increases with highly correlated stimulus (Figure 3b). On the other hand, decoding performance *D* increases with higher stimulus correlation (Figure 3c). This trend of increased decoding performance is similar to the increase in asymmetry of the network. Lastly, the relative decoding performance compared to independent neural population $D^{*}/D_{ind}$ increases in general as the stimulus correlation increases (Figure 3d).

We alter the temporal correlation of dynamic input by tuning the diagonal value of matrix *M* in the stochastic Equation (Equation 12). This is effectively the inverse time scale of autocorrelation $\langle x_{i}\left(t^{\prime }\right)x_{i}\left(t\right)\rangle \propto \exp\left(\frac{-|t^{\prime }-t|}{\alpha }\right)$. The result shows that the asymmetry η of optimal network increases along with the temporal correlation scale α (Figure 4a). The entropy production is maximized at an intermediate time scale (Figure 4b). The decoding performance *D* increases with a similar trend as η (Figure 4c). The relative performance versus an independent population $D^{*}/D_{ind}$ also increases when the stimulus temporal correlation is longer (Figure 4d). Together, we show that optimal network connectivity relates to the statistical structure of dynamical input, increasing asymmetry and entropy production when stimulus is correlated, and the decoding performance always outperforms independent neural populations.

### 3.3. Optimal Population Coding and Network Properties

We alter reliability of the network by tuning the inverse temperature β in the kinetic Ising model (Figure 5). In other words, β controls the reliability of neural response in the network. The result shows that asymmetry η of optimal network is not monotonically related to the response reliability (Figure 5a). Asymmetry peaks near the critical temperature of kinetic Ising models. Entropy production increases and saturates as the reliability increases (Figure 5b). The decoding performance *D* follows a similar trend (Figure 5c). Lastly, decoding compared to independent population $D^{*}/D_{ind}$ does not show a specific trend against network reliability, but note that the values are all larger than one (Figure 5d).

We further analyze how the reliability of optimal network relates to the structure of population code (Figure 6). The result recovers previous findings, showing that the network effectively decorrelates stimulus at high reliability and forms redundant population code at lower reliability [30]. This is shown by plotting the covariance of input projection Φx against the network connectivity Figure 6. The relation qualitatively holds under the framework with nonequilibrium networks. Together, this shows that the reliability of neurons, or effective temperature of the kinetic Ising model, affects the decoding performance and coding strategy of optimal networks.

Furthermore, we investigate how network properties scale with input dimension *d* or network size *N*. We measure entropy production with input signal with different dimensions. The result shows that entropy production remains small for low dimensions but may increase above a certain dimension (Figure 7a). We also measure decoding performance with an increasing number of sub-population of neurons and show that it gradually increases and slightly saturates (Figure 7b). This agrees with the redundant population coding structure empirically discovered [14,30].

### 3.4. Generalization to Recurrent Networks and Nonlinear Stimuli

We generalize the discrete settings in the kinetic Ising model to a continuous recurrent neural network (RNN) framework, then analyze neural trajectories and decoding performance. A recent method trains chaotic RNNs through error feedback from the target signal as well as another driven network [17]. The iterative least square learning algorithm modifies network weights to generate target dynamics. Our approach is similar, effectively using a step-wise procedure. We generate spiking patterns for optimal linear decoding as a target, then reconstruct an effective connectivity that is trained to generate such spiking patterns (Figure 1 and Figure 2). The qualitative agreement of the two approaches would support the idea that spiking and rate neural networks may have similar properties in their network structures (Figure 8).

Specifically, the RNN has a linear readout similar to Equation (Equation 1) from rate activity r. The network receives feedback error signal and the iterative least square learning algorithm follows:(18)$$J\left(t\right)=J\left(t-\Delta \right)-e{\left(t\right)}^{T}P\left(t\right)r\left(t\right)$$
(19)$$P\left(t\right)=P\left(t-\Delta \right)-\frac{P\left(t-\Delta \right)r\left(t\right)r{\left(t\right)}^{T}P\left(t-\Delta \right)}{1+r{\left(t\right)}^{T}P\left(t-\Delta \right)r\left(t\right)}$$
where *e* is the error term between the ongoing signals and targets, *P* is the inverse correlation matrix ${\langle r\left(t\right)r{\left(t\right)}^{T}\rangle }^{-1}$ approximated online, r is the firing rate, and Δ is the discrete time step. We show that asymmetry of the network increases with time scale of the stimuli (Figure 8c). This agrees with the known calculation in kinetic Ising models, where asymmetry of the pairwise interaction contributes to entropy production. Spectral analysis of the network also reflects that the optimal coding spectrum tends to match the frequency mode of dynamic input (Figure 8d). The network mode is computed through eigen-decomposition of the connectivity matrix: λu=λJ and the inverse of the larges eigenvalue 1/|λ| characterizes the dominating time scale.

Dynamic input applied to the network in the past sections is described by linear stochastic equations in the form of Equation (Equation 12). We explore nonlinear dynamics as stimuli with underlying low dimensional structures such as bistability. Specifically, the dynamical stimuli have a mixture of two Gaussians, one with positive and one with negative mean values, and variance controls the apparent transition between two states (Figure 9a,b). This can be constructed with a state-switching control signal in Equation (Equation 13). At the limit of infinite time, the correlation and autocorrelation can be similar for two time series with different bistability. The optimal network indeed reflects the stimulus statistics and decreases both asymmetry entropy production as the overlap of two states increases (Figure 9c,d). This is measured by the probability of observing values in between two states, corresponding to the exponential of a barrier in the effective energy landscape.

With two mixtures fused into one state, the optimal network is close to symmetric with little entropy production, congruent with the result of Hebbian learning for single memory pattern. In contrast, under stimuli with periodic transitions between two distinct states, the connectivity learns both attractor states and the transition between them. The dwell time between these basins reflects the transition rate of stimuli. Together, these physical characteristics relate network dynamics with nonequilibrium physics and show how optimal networks should match properties of the input dynamics.

## 4. Discussion

We study nonequilibrium network parameters that generate optimal population code for dynamic stimuli. We formulate the problem with the kinetic Ising model and leverage its inference methods. The inference procedure is used to learn network parameters that generate neural activity for stimulus decoding. The result shows how network asymmetry, decoding performance, and entropy production depend on stimulus statistics. The reliability of neural response can change the decoding performance as well as the encoding strategy. The results further generalize to continuous time networks and encoding of nonlinear input. This provides insight into the design principle of that receive and encode dynamical signals and its relation to nonequilibrium physics.

Previous literature tends to study stimulus encoding for ensembles of static input “frames” [30,39]. For simplified networks like the Ising model, the focus has mostly been on equilibrium properties [30,32]. An extension of the MaxEnt framework is to study the response pattern conditioned on stimulus, which shows more robust prediction in spiking patterns and quantifies information encoded in neural populations [11]. However, these methods are still applied in an equilibrium setting without considering input dynamics. Recent literature extends theories of efficient coding in single channel or a population by adding adaption, which adjusts parameters of the encoder depending on the non-stationary input, environmental context, and loss function parameterized by a given task [8,40,41,42]. The difference compared to this work is that the neural population is mostly non-interacting or forming a feed-forward network, whereas we focus on the recurrent structure. In addition to the classical efficient coding, other paradigms such as predictive coding that incorporates an objective to predict spatiotemporal features and sparse coding that aims for a sparse representation have been integrated in an information theoretical framework [43]. Adding these coding principles as constraints to the readout or network activity in our setup can be explored in the future.

Our findings agree with encoding in equilibrium models when input has few temporal feature, but the investigation further generalizes to dynamical stimuli. We show that the asymmetric network always outperforms uncorrelated populations, and the difference increases along with larger stimulus correlation (Figure 3 and Figure 4). This reflects the functional benefit for asymmetric structure if the network learns to encode signal with spatiotemporal correlations. The intuition can come from the learning rule in Equation (Equation 9). When the gradient vanishes, the steady-state solution is matching the time delayed correlation ${\langle s_{i}\left(t+1\right)s_{j}\left(t\right)\rangle }_{t}$ with an approximated correlation between the effective field and current spiking pattern ${\langle \tanh\left(\beta H_{i}\left(t\right)\right)s_{j}\left(t\right)\rangle }_{t}$. The connectivity is therefore trained to capture the temporal structure through comparing the current input and consecutive spike pattern. For stimulus correlation, this is reflected in the dimensionality of network projection that affects the effective field *H*. The network can pickup the temporal correlation more easily as the stimulus has lower effective dimensions.

Another part of our result is related to how the reliability of neural responses affect stimulus encoding in the network (Figure 5). A similar study has been done on the MaxEnt model, where the encoding strategy is more redundant at low reliability and uses decorrelation at high reliability [30] (Figure 6). This result qualitatively holds for asymmetric network and indicates that connectivity should account for robustness of neural response to produce optimal population code. Importantly, asymmetry seems to peak at a certain reliability, namely the inverse of a critical temperature, and may be related to physical properties of the kinetic Ising models [25]. The functional benefit for signal processing in the nervous system is a popular hypothesis that continues to be tested empirically and theoretically [14,44]. Fluctuation and correlation across units in the network increases near the critical point ($\beta_{c}∼1.11$ for kinetic Ising model [35]). For a given kinetic Ising model, entropy production maximizes near the critical point. Note that our comparison of the nonequilibrium measurements is with respect to an optimal network trained to encode dynamical signals. This can potentially explain the results in Figure 5.

Recently, nonequilibrium physics has been applied to analysis of network dynamics. The results show how nonequilibrium measurements such as entropy production, nonequilibrium work relation [21], nonequilibrium free-energy [45], and heat dissipation can be characterized in neural networks [46]. Specifically, dissipated heat is minimized during network training in the restricted Boltzmann machine [46]. In another study, optimal learning rule for neural connectivity can be characterized by efficiency formulated with thermodynamics measurements [47]. Theoretical work in kinetic Ising model explores entropy production and nonequilibrium phase transitions [25]. The non-monotonic results of network encoding for different effective temperature may draw links with critical behavior of kinetic Ising model [35]. Future studies can focus on the relation between these nonequilibrium phenomena and decoding performance. Specifically, investigation of functional benefits near the nonequilibrium critical point is still needed.

Recurrent networks have been trained to encode or generate dynamical signal in the past few decades [16,17,48,49]. The original works started from echo-state and liquid-state machines with interacting nonlinear units that receive continuous dynamical input, and the linear readout weight is adjusted to match the target signal [50]. This differs from the setting in this work, where the internal network is learned and the network has binary activity. However, features shown near criticality seem to agree with findings from echo-state machines, where the optimal decoding performance exists near the critical point of the dynamical system. The edge of critical point in echo-state machine is where the system enters a chaotic regime, whereas the critical temperature in kinetic Ising model is an nonequilibrium analogue of ferromagnetic phase transition [35]. An extension to training continuous networks with chaotic dynamics is known as the FORCE algorithm. In these training procedures, the network starts with a larger population of randomly connected neurons and the learning process updates the connectivity according to a feedback error signal through an iterative least square algorithm [16,17]. This approach relies on a random network that is sufficiently large and produces chaotic dynamics. It is unclear if the initial condition of random connections play a significant role in the trained network. Recent studies focus on low-rank networks with well defined input and target signals [18]. In these settings, stimulus encoding of the network is built-in according to the desired signal. Optimal performance of these low-rank networks depends on temporal structure of the dynamic stimuli [31]. These findings agree with our results showing that the optimal network structure reflects the statistics of dynamical input. Specifically, we showed that both the network asymmetry and its dominating time scale positively correlate with the temporal correlation of target signals. This result qualitatively agrees with the kinetic Ising model, despite using different learning algorithm and evolving with continuous dynamics (Figure 8). Lastly, we extend the measurements beyond linear stimulus statistics and show that bistability can also be reflected in the nonequilibrium measurements of optimal networks (Figure 9). The results can be compared to studies using landscape theory to characterize nonequilibrium networks, showing how transition kinetics between basins of the energy landscape are affected by network asymmetry [22]. Here we are not tuning network asymmetry as a parameter but altering the statistics of target dynamics. The result is consistent in showing that asymmetry is related to nonequilibrium flux between activity patterns, and here the activity patterns reflect the target dynamics it is trained to encode.

Another thread of study is the neural engineering framework (NEF), which constructs a structured network that encodes and decodes dynamical systems. The way we generate target spike trains for optimal linear decoding is similar to NEF. However, NEF differs from our approach as it optimizes the optimal decoding matrix with knowledge of equations of the dynamical stimuli, rather than training the network in our setting [48]. Finally, another related framework called “neural engine” has been used to draw a link between the Bayesian brain hypothesis and energy-based models [23]. The network response to external stimuli can be viewed as a thermodynamic process, similar to a heat engine, and the efficiency is defined in terms of entropy emitted from stimulus response. The formulation of the posterior response in Equation (Equation 3) was inspired by the Bayesian brain framework for stimulus coding [3,23]. By computing the mode of the distribution in Equation (Equation 3), the spike generating process is similar to the classic linear-nonlinear model that passes the filtered input through a sigmoid nonlinearity in Equation (Equation 4). Note that the presented algorithm is not fully Bayesian, since the spike generation is from a fixed independent prior for neural activity. This is not updated according to the learned network structure and potentially correlated neural activity. In addition, the spike generation process Equation (Equation 4) uses the mode rather than the full spike rate distribution for simplicity. Future studies can introduce iterative learning process to optimize the objective in Equation (Equation 5) with updates in the spiking process.

In addition to continuous neural dynamics, modern machine learning studies introduce algorithms for training spiking RNNs [48,49]. The analysis of population code and nonequilibrium measurements introduced in this work can be applied to these frameworks as well, potentially enabling one to understand the trained network mechanisms. The kinetic Ising model is obviously a much simpler framework as a spiking RNN, and the learning algorithms have already been explored in the past [12,34]. However, the benefit of analyzing population coding in this setup is that the optimality is relatively well defined given the linear decoding setup and network inference formula. The training process is also simpler, without more hyperparameters or network architecture to be selected. Lastly, the thermodynamical interpretation can be understood and compared to Ising models under the MaxEnt method [44,51]. As shown in the extension of chaotic rate neural networks, we expect future works to investigate nonequilibrium properties of more complex spiking networks.

Modern neuroscience technologies developed methods to record massive amount of neural activities in parallel, offering challenges and opportunity for analysis of high dimensional data [52]. A challenge is to characterize the functional connectivity of neural networks from time series of neural activity [52,53,54]. Classic methods such as GLM and MaxEnt characterize the neural population interaction in terms of coupling kernels and pair-wise interaction, respectively. Recent studies derive GLM-like formula for a network of leaky-integrate and fire neurons, while constructing an effective potential that has similar interpretation of MaxEnt methods [44]. The general relation between statistical models and the thermodynamic formalism of spike trains is still less clear. Here, the kinetic Ising model framework has inference similar to maximum likelihood in GLM framework and physical interpretation related to physical models using MaxEnt. The target spike train for decoding dynamic signal is generated through Bernoulli GLM, while the encoder is trained with a kinetic Ising model. This series of computational work can potentially mitigate the gap between statistical and physical models for neural coding. In addition, analytic study proved exact mapping between kinetic Ising model and autoregressive models with certain constraints on the network parameters, suggesting shared inference method in state-space models [55]. For the training procedure, the learning rule based on likelihood of the spin update is similar to a form of spike-time dependent plasticity (STDP), where correlation of the pre- and post-synaptic neuron is compared with the evoked potential [44,56]. A full form of STDP rule was derived through the free-energy of Boltzmann networks or networks with hidden units [56,57]. The comparison with the simplified form in our work should be explored in the future. On the other hand, mean field methods have been used to optimize the inverse problem of Ising models, and recent theoretical works extend it to nonequilibrium systems [27,35,37]. Different approximations can be unified in an information geometrical framework [35]. We applied a subset of mean field methods in our studies and the results were consistent, but a systematic comparison would be beyond the scope of this work.

While we simplify neural dynamics as kinetic Ising models, similar analysis can be applied to real experiments in systems neuroscience. For instance, in sensory systems, one can study the encoding performance of different dynamical stimuli and analyze the effective nonequilibrium connectivity during stimuli. Parameters such as asymmetry of the network, decoding performance, and entropy production can be compared to our numerical work. Similar analysis can be performed in motor systems given different dynamics of the target motor output [18]. One can measure an asymmetric effective connectivity given the temporal dynamics of the target output [58,59]. In addition, in certain model systems, given the ground truth neural connectivity, one can predict the optimal encoding and decoding dynamical signal [54]. Empirically, we might have to consider the dynamics of synaptic connection as well as the effects of learning on longer time scales. With a simplified neural model, we note that there are a number of assumptions in this framework, including binary spiking patterns, the instantaneous pair-wise interaction, and linear decoding for target dynamics. More elaborate models with biophysical details can be studied in the future.

## Figures and Tables

**Figure 1 entropy-24-00598-f001:**
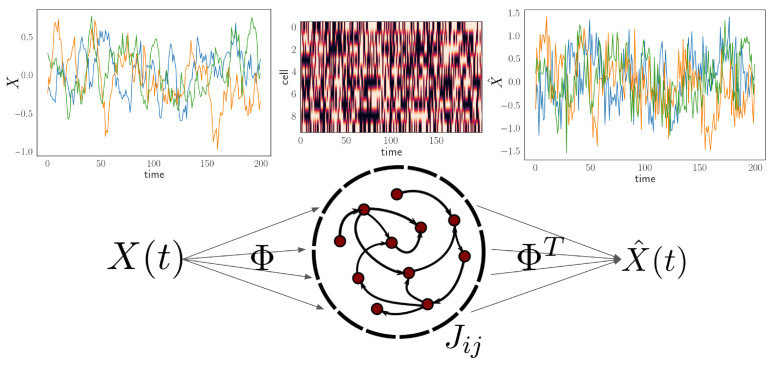
Setting for encoding dynamic stimuli with a network based on kinetic Ising model. The external stimuli X(t) start from the left hand side and the example time series shown above. The stimulus with dimension d=3 is projected to the network N=10 with connection $J_{ij}$ through Φ. Activity of the neural population is shown above the network schematic and note that arrows between the network nodes indicate directed connections. The output $\hat{X}\left(t\right)$ is reconstructed through projection $\Phi^{T}$ on the right side and the example time series is shown above. The goal of this setting is to find optimal network parameters $J_{ij}$ to reconstruct stimuli X(t).

**Figure 2 entropy-24-00598-f002:**
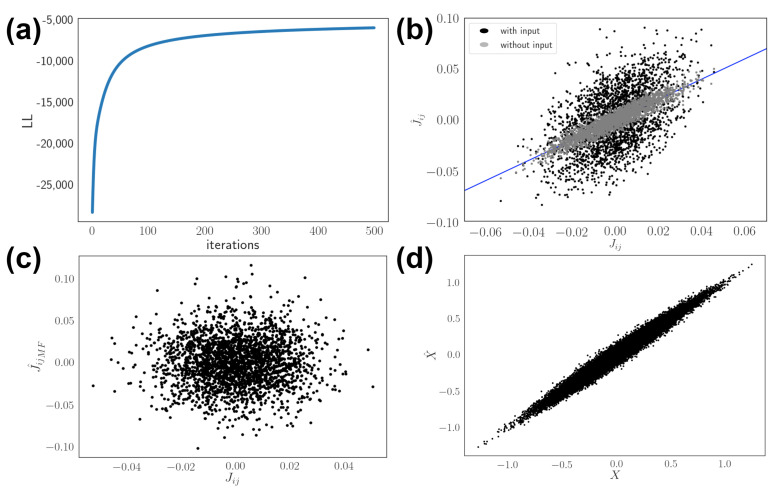
Network inference for kinetic Ising model and stimulus decoding. (**a**) Maximizing log-likelihood (LL) through gradient method. (**b**) The result for maximum likelihood method shows agreement between inferred ${\hat{J}}_{ij}$ and true $J_{ij}$ parameters. Estimation for network driven by input (black) is more biased and scattered compared to network without input (grey). Diagonal blue line shows an exact match. (**c**) The result of mean field ${\hat{J}}_{ijMF}$ approximation method. (**d**) Stimulus decoding under the inferred optimal neural network parameter through maximum likelihood. Here the network size is larger for visualization N=50.

**Figure 3 entropy-24-00598-f003:**
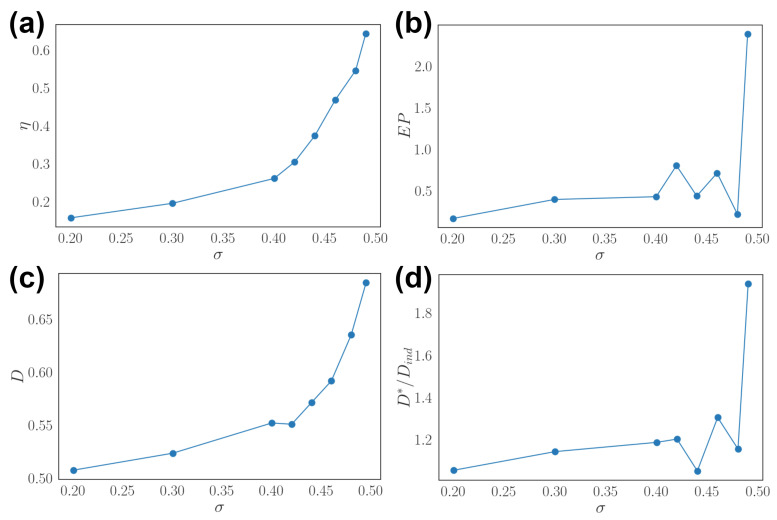
Effects of stimulus correlation. (**a**) Asymmetry η as a function of stimulus correlation $\sigma =\langle xx^{T}\rangle$. (**b**) Entropy production (EP) as a function of σ. (**c**) Stimulus decoding *D* as a function of σ. (**d**) Relative decoding performance $D^{*}/D_{ind}$ as a function of σ.

**Figure 4 entropy-24-00598-f004:**
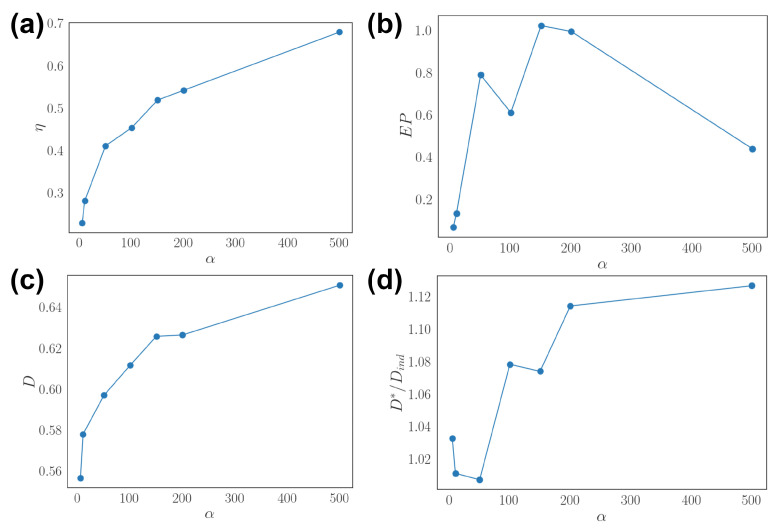
Effects of temporal correlation of dynamical stimuli. (**a**) Asymmetry η as a function of temporal correlation scale α. (**b**) Entropy production (EP) as a function of α. (**c**) Stimulus decoding *D* as a function of α. (**d**) Relative decoding performance $D^{*}/D_{ind}$ as a function of α.

**Figure 5 entropy-24-00598-f005:**
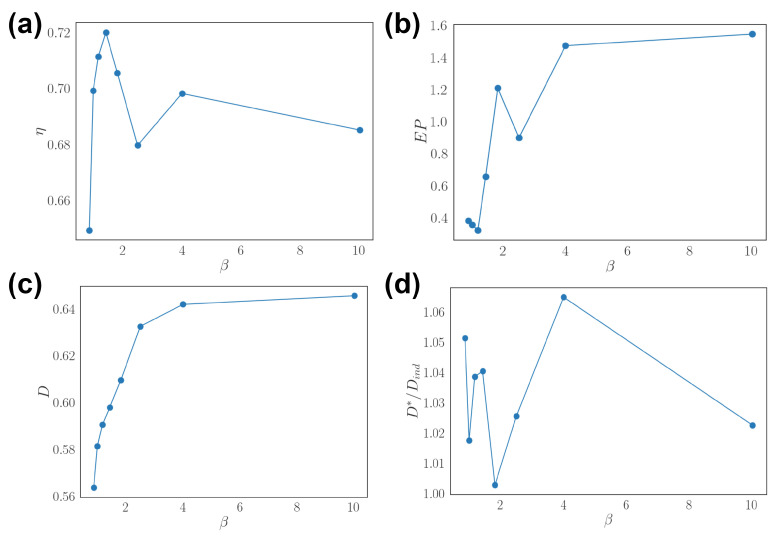
Effects of network response reliability. (**a**) Asymmetry η as a function of network reliability β. (**b**) Entropy production (EP) as a function of β. (**c**) Stimulus decorrelation *D* as a function of β. (**d**) Relative decoding performance $D^{*}/D_{ind}$ as a function of β.

**Figure 6 entropy-24-00598-f006:**
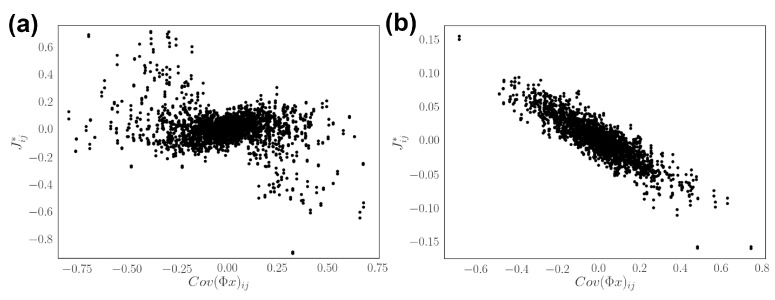
Encoding strategies of networks with different response reliability. (**a**) Input stimulus correlation versus the optimal connectivity under low reliability (β=1). (**b**) Input stimulus correlation versus the optimal connectivity under high reliability (β=10). Here we use a larger network N=50.

**Figure 7 entropy-24-00598-f007:**
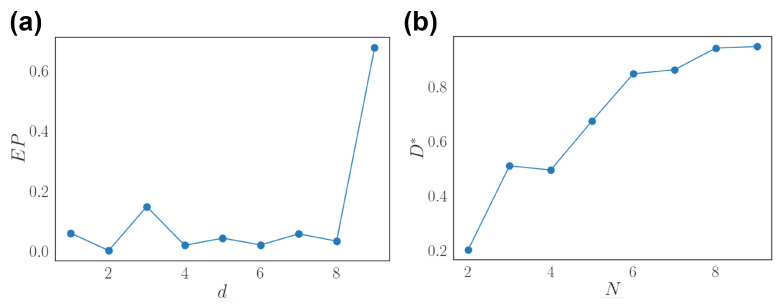
Scaling properties of the nonequilibrium network. (**a**) Entropy production (EP) as a function of the input dimension *d*. (**b**) Optimal decoding performance $D^{*}$ as a function of the number of neurons considered *N*.

**Figure 8 entropy-24-00598-f008:**
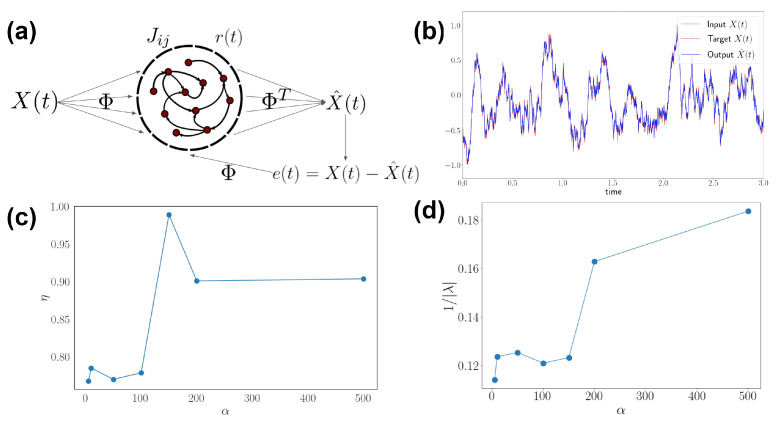
Properties of continuous recurrent neural networks. (**a**) The schematic of a framework with continuous recurrent networks and training for optimal network encoding. Note that this is similar to the setup in Figure 1, but with rate variables r(t) and training through error signal e(t). (**b**) The performance of stimulus decoding with the trained optimal network. One-dimensional (d=1) time series show input signal (green), target (red), and output (blue) of the network. Input and target are the same for our stimulus encoding study. Result is for N=100 network with 10 trials following full-FORCE method according to [17]. (**c**) Asymmetry of the network as a function of stimulus correlation α. (**d**) Dominating time scale (1/|λ|) as a function of stimulus correlation.

**Figure 9 entropy-24-00598-f009:**
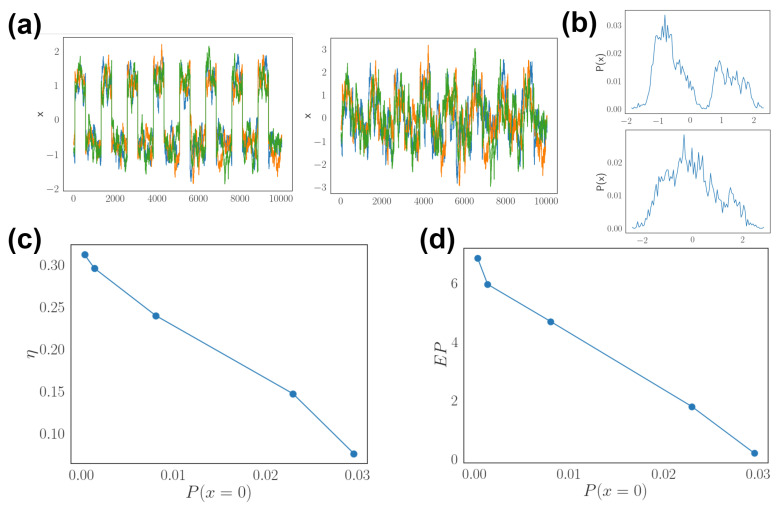
Effects of bistable dynamic input. (**a**) Time series of highly separated states (**left**) and overlapping states (**right**) as the bistable input pattern x. (**b**) The probability distribution of input values P(x) corresponding to two patterns, with the upper panel from left and lower panel from right patterns in (**a**), with P(x=0)=0.001 and P(x=0)=0.023, respectively. (**c**) Asymmetry η as a function of the probability of observing x=0, which is the unstable transition point between two states. (**d**) Entropy production as a function of P(x=0).

## Data Availability

Scripts and demo for analysis and the network model would be available: https://github.com/Kevin-Sean-Chen/OptNE.

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
