# Peer review of "Optimal Population Coding for Dynamic Input by Nonequilibrium Networks"

_entropy, 2022, doi:10.3390/e24050598_

Round 1

Reviewer 1 Report

In this study, the Author optimises connectivity of kinetic Ising models for optimal linear decoding of time-series. It then explores how the properties of these optimal networks (decoding accuracy, asymmetry and entropy production) depend on stimulus statistics and reliability of individual neurons. Finally, the paper discusses how recurrent, continuous (rate-based) neural networks behave in similar parameter regimes. The paper thus explores relationships between various research directions in neural coding and dynamics. 

Overall, I find the addressed topics interesting and the scope of the paper to be broad. I agree with the Author that there is a clear need for more rigorous, theoretical approaches to neural coding, which analyse dynamic stimuli and population effects. At the same time, I think that the paper could be significantly improved along four directions which I discuss in more depth below (see major concerns).

Major concerns:

  1. Statement of the problem and relevance of the results. The paper synthesises a number of approaches - from neural coding to non-equilibrium statistical physics. While this clearly implies the breadth of the Author’s interests, the central motivation of the study is unclear. For example, the introduction speaks about relationship of GLMs and MaxEnt approaches, while the paper does not revisit this question at any depth. The paper does not articulate a clear problem which needs addressing. The last paragraph of the Introduction  (lines 69-75) summarises the results, but does not explain how this particular study contribute to the field beyond reporting on simulation results. I strongly suggest restating the problem better, distilling the key results in a concise form and discussing the contributions of the paper in either Introduction or Discussion. 
  2. Literature and context of the study. The paper seems to omit a number of relevant results in the field, which impacts the interpretation and positioning of results. For example, the Introduction states that “However, prior literature mostly focus on single neuron coding, non-interacting population, or with static stimuli” (lines 22-24). These issues were and continue to be subjects of active research, and a number of studies addressed them in recent years (see some example references below). My suggestion is to discuss relationship to recent published results much more in depth.
  3. Mathematical aspects and analysis. I find the mathematical notation as well as terminology are somewhat confusing. In particular:
    1. Eq. 2 - what is s on the LHS - should it be r? Also - if \hat{x} is a linear decoding of the stimulus, with the assumption of Gaussian uncertainty then the probability distribution is over x i.e. p(x|r) = N(\hat{x}, \Sigma)
    2. Eq. 3 - \Omega is never referred to again - where does it come from? If the prior over r is flat (as stated later), why the need of stating the prior parametrised by \Omega? Overall - the presentation of the model seems very confusing. Eg. why is a Bayesian inversion required to compute the encoding distribution? This distribution typically reflects the mechanics of the encoding model. Where does the Ising like prior come from and why is it a good choice? If r is continuous, then what does it mean to have an Ising-like prior over a continuous vector - isn’t that just a multivariate Gaussian? The meaning of individual terms in eq. 3 could be also better clarified in the text.
    3. Related to the above -  eq.3 and eq.4 - \lambda(t) is “the spiking rate from equation 3” - what does this mean? Does it mean that “\lambda(t) = r”? but then - where does the time dependence enter? Also (perhaps most importantly) - how are the spike counts s(t) generated from the *distribution* p(r|x)? Eq. 4 specifies how spikes are generated at an individual time instant from a scalar firing rate \lambda. Eq. 3 specifies a static distribution over *vectors* r. The paragraph refers to s(t) as “spike counts”, but per Eq. 4 s(t) is a binary variable. This should be explained and mathematical notation should be made more clear.
    4. Why this additional step (i.e. eq 4) for spike generation is required ? A probabilistic encoding model should specify an explicit probability distribution over the responses given the stimuli i.e. p(r|x) or p(s|x). If for some reason the two-step procedure is necessary (first p(r|x) and then p(s|r)), then it should be clearly explained. It should be also said why there is no statement of a single distribution p(s|x). 
    5. Eqs 9-10 - what is the gradient exactly with respect to? Is it eq. 3 (the encoding distribution)?Or is it equation 7? But then - what is optimised? And where does the dependency on the stimulus enter?
    6. Quality assessment (eg. in Fig 2b) - it seems that the algorithm does not recover the ground-truth connectivity very well - there is a number of values which differ not only in magnitude, but also in sign. it is very hard to judge how well does the algorithm perform without a quantitative measure of performance. Also - how is correlation D computed over multidimensional inputs?
  4. Parameter ranges. The simulations are reported (with few exceptions) for a single set of parameter values (described on page 5). How robust are the results with respect to parameter settings? This should be addressed in some form to support the generality of the claims.

Minor concerns:

  1. Lines 108-110: HMM model with observed states is a Markov model. There seems to be no need to note the similarity.
  2. Fig 3c - results are very counter-intuitive - why does the decoding accuracy decrease with increased stimulus correlation?  The initial expectation is that more redundant stimuli are more easily decodable - isn’t that correct?
  3. This is not a concern - just a remark - the setting seems somewhat related to Liquid State Machines and Echo State Networks. There however one typically adjusts only the decoding weights. Perhaps it is worth discussing the relationship?

Typos:

Line 311: peek -> peak

Literature suggestions (these are just a few recent examples):

Röth, Kai, Shuai Shao, and Julijana Gjorgjieva. "Efficient population coding depends on stimulus convergence and source of noise." PLoS computational biology 17.4 (2021): e1008897.

Berkowitz, John A., and Tatyana O. Sharpee. "Quantifying information conveyed by large neuronal populations." Neural computation 31.6 (2019): 1015-1047.

Bellec, Guillaume, et al. "Long short-term memory and learning-to-learn in networks of spiking neurons." Advances in neural information processing systems 31 (2018).

Chalk, Matthew, Olivier Marre, and Gašper Tkačik. "Toward a unified theory of efficient, predictive, and sparse coding." Proceedings of the National Academy of Sciences 115.1 (2018): 186-191.

Mlynarski, Wiktor F., and Ann M. Hermundstad. "Efficient and adaptive sensory codes." Nature Neuroscience 24.7 (2021): 998-1009.

Granot-Atedgi, Einat, et al. "Stimulus-dependent maximum entropy models of neural population codes." PLoS computational biology 9.3 (2013): e1002922.

Weber, Alison I., Kamesh Krishnamurthy, and Adrienne L. Fairhall. "Coding principles in adaptation." Annual review of vision science 5 (2019): 427-449.

Author Response

Dear Reviewer,

Please see the attachment for reply to review report. Thanks you for the constructive feedback!

Reviewer 2 Report

The article gives a surround overview and introduction into Ising-model-based network learning. It reviews different approaches in the literature and analyzes the behaviour of the Ising network with respect to nonequilibrium inputs. It investigates the structure of stored links dependent on the external input. Lower effective dimensions result into larger asymmetry. The temporal correlation scale is found to increase the asymmetry as well. In this respect the sentence that the decoding performance always outperforms independent neural populations could be explained more in detail. The dependence of the response liability on the inverse temperature in figure 5d is quite astonishing. Though all other parameters in the figure like the entropy increase shows a monotonous and understandable behaviour the relative decoding performance shows a minimum followed by a maximum in temperature. This is rather exciting and deserves more focus in the discussion. The article investigates extension by positive feedback to optimize the learning matrix with remarkable results. In Figure 9 two different inputs of separated and overlapping states are analyzed. One would expect two lines as output to be compared in 9c,d. This is perhaps a misunderstanding. Overall the paper should be published with the optional suggestions to extend the discussions at the above mentioned places.   

Author Response

(The authors gave the same response as above.)

Reviewer 3 Report

Considering Eq. (9), I think the learning rule, introduced in the paper, is a reduced version of the STDP rule although the author did not mention about it.  The frame and ability of the learning rule, such as asymmetric change in connectivities depending on firing order, processing of temporal features, are the same with those of the STDP rule.  It was also suggested that the STDP rule can be derived from the gradient descent on the free energy, instead of the negative of the log probability.

1. I notice that a network is trained to generate the output \hat{X}(t) being the same with the input X(t), but I cannot know under what condition a trained network reconstructs or generates the output. The learning goal would be nonsense If the trained network generates the output only during it receives the input X(t).

2. According to Fig.1, a trained network seems to have the ability to generate different outputs depending on conditions.   I wonder whether a network really can be trained to learn several temporal patterns and generate one of them depending on a given  condition. Otherwise, Fig.1 should be revised.

3. In the issues of time series modeling and prediction, it is known that a time series can be modeled easily or not depending on several characteristics of the time series, such as nonlinearity, correlation demension, optimal time delay for atractor reconstruction, Lyapunov exponents.  And, the time series generated by the linear stochastic equation may have predictable characteristics.  I wonder whether the learning rule works properly for a time series with different characteristics.

4. I cannot figure out how the entries of \Phi are given and the role of \Phi.

Minor opinion

- Some mathematical expressions are so ambiguous. I recommend at least vectors and matrices are expressed in a more distinct way.

Author Response

(The authors gave the same response as above.)

Round 2

Reviewer 1 Report

The Author has responded to my review. Unfortunately, I have to say that I have a hard time understanding the response. No actual improvements of the mathematical work were made, beyond some textual modifications. As it currently stands, I do not understand what is the paper trying to achieve, nor the means by which it is trying to achieve its goals.

Given this lack of clarity, the somewhat convoluted and non-standard use of the terminology, as well as the fact that the Author admits that in Fig 3C in the first version a major mistake was made casually (it was demonstrating as if the increasing stimulus correlations decrease the decoding accuracy) I can not endorse publication of the paper in its current form.

Author Response

We thank the reviewer for the informative comments. Please check the attached file for response to this revision. Thank you!

Reviewer 3 Report

..

Author Response

We thank the reviewer for constructive feedback in the last review report. In this round of revision, according to suggestions from other reviewers, we continue to edit the mathematical formulism and emphasis on the contribution of this work in the introduction and discussion sections. We hope this brings clarity to the work. We thank the reviewer again for reviewing and endorsing this work.

This manuscript is a resubmission of an earlier submission. The following is a list of the peer review reports and author responses from that submission.